# Shared Pathogenic and Therapeutic Characteristics of Endometriosis, Adenomyosis, and Endometrial Cancer: A Comprehensive Literature Review

**DOI:** 10.3390/ph17030311

**Published:** 2024-02-28

**Authors:** Melinda-Ildiko Mitranovici, Diana Maria Chiorean, Liviu Moraru, Raluca Moraru, Laura Caravia, Andreea Taisia Tiron, Titiana Cornelia Cotoi, Havva Serap Toru, Ovidiu Simion Cotoi

**Affiliations:** 1Department of Obstetrics and Gynecology, Emergency County Hospital Hunedoara, 14 Victoriei Street, 331057 Hunedoara, Romania; 2Department of Pathology, County Clinical Hospital of Targu Mures, 540072 Targu Mures, Romania; chioreandianamaria@yahoo.com (D.M.C.); ovidiu.cotoi@umfst.ro (O.S.C.); 3Department of Pathophysiology, “George Emil Palade” University of Medicine, Pharmacy, Science, and Technology of Targu Mures, 38 Gheorghe Marinescu Street, 540142 Targu Mures, Romania; 4Department of Anatomy, “George Emil Palade” University of Medicine, Pharmacy, Sciences and Technology, 540142 Targu Mures, Romania; liviu.moraru@umfst.ro; 5Faculty of Medicine, “George Emil Palade” University of Medicine, Pharmacy, Sciences and Technology, 540142 Targu Mures, Romania; raluca.moraru@umfst.ro; 6Division of Cellular and Molecular Biology and Histology, Department of Morphological Sciences, “Carol Davila” University of Medicine and Pharmacy, 050474 Bucharest, Romania; laura.caravia@umfcd.ro; 7Faculty of Medicine, “Carol Davila” University of Medicine and Pharmacy, 050474 Bucharest, Romania; taisia_andreea@yahoo.com; 8Department of Pharmaceutical Technology, “George Emil Palade” University of Medicine, Pharmacy, Sciences and Technology, 540142 Targu Mures, Romania; titiana.cotoi@umfst.ro; 9Close Circuit Pharmacy of County Clinical Hospital of Targu Mures, 540072 Targu Mures, Romania; 10Department of Pathology, Akdeniz University School of Medicine, Antalya Pınarbaşı, Konyaaltı, Antalya 07070, Turkey; seraptoru@akdeniz.edu.tr

**Keywords:** endometriosis, adenomyosis, pathogenesis, endometrial cancer, malignant behavior, personalized treatments

## Abstract

Endometriosis and adenomyosis behave similarly to cancer. No current treatments represent a cure, even if there are several options, including hormonal and surgical therapy. In advanced or recurrent pathologies, however, personalized treatment is necessary. We have found that due to the multiple common features, various therapeutic options have been used or studied for all three pathologies, with varying results. The objective of this review is to extract from the relevant literature the compounds that are used for endometriosis and adenomyosis characterized by malignant behavior, with some of these drugs being studied first in the treatment of endometrial cancer. Special attention is needed in the pathogenesis of these pathologies. Despite the multiple drugs that have been tested, only a few of them have been introduced into clinical practice. An unmet need is the cure of these diseases. Long-time treatment is necessary because symptoms persist, and surgery is often followed by postoperative recurrence. We emphasize the need for new, effective, long-term treatments based on pathogeny while considering their adverse effects.

## 1. Introduction

Endometriosis and adenomyosis exhibit characteristics that closely resemble those of cancer, leading to the adaptation of many cancer treatments to manage these conditions. To comprehend this similarity, it is essential to focus on understanding the intricate pathogenesis of these diseases. Endometriosis involves the presence of endometrial glands and stromal cells outside the uterus, while adenomyosis involves these cells growing within the myometrium. The complex origins of these diseases are still debated and bear resemblances to cancer’s pathogenesis, significantly affecting the quality of life of patients living with them [1,2,3].

For now, there is no gold standard regarding the treatment of this disease, while there have been advances in the imaging criteria for diagnosing endometriosis and adenomyosis in order to avoid surgery [2,3]. There are no specific guidelines to follow for the best management. Medical treatment is based on the pathogenic mechanisms [1,2]. None of the current treatments represent a cure, even if there are several options, which include hormonal and surgical therapies. Histological and genetic alterations in the endometrium lead to changes associated with several types of cancers. Malignant transformation is rare, and in general, the prognosis is good. The exact mechanisms of these transformations, however, are still unknown [1]. In general, the consequences of endometriosis or adenomyosis are often under-reported because of their heterogeneity and the delays in diagnosis [1,4].

Endometriosis affects 10–15% of all women of reproductive age. To date, the accepted etiopathogenesis is retrograde menstruation and transplantation of shed endometrium. The development of endometriosis is still being researched [1]. Adenomyosis is another disease that affects the quality of life of those living with it, similarly to endometriosis. The reported incidence varies between 5% and 70%. This fact suggests that the occurrence of adenomyosis is not well known, but it mainly affects 35–50-year-old women [5].

A continuous increase in the incidence of EC (endometrial cancer) has been observed. In highly developed countries, this is probably due to the tendency toward late motherhood [6]. EC is the most common gynecological cancer in the USA, with its incidence rising [6,7]. Patients with advanced and recurrent EC are a major therapeutic challenge [8]. The severe prognosis of this disease in comparison with endometriosis and adenomyosis clarify the major efforts of researcher to find more effective therapeutic options. Also, the treatment aim of endometrial cancer is different from endometriosis and adenomyosis [8,9].

While endometrial cancer shares some pathogenic features with endometriosis and adenomyosis pathways, it is critical to emphasize the fundamental difference between these diseases, which consists of the severity of the endometrial cancer. The main concern in this case is mortality and the treatment consists of surgery with or without adjuvant therapy, personalized treatment being applied in the case of recurrence or lack of response to treatment. We have to recognize this critical distinction between these pathologies before exploring various shared therapeutic options based on the complex nature of their pathogenesis.

Notably, medications initially developed for endometrial cancer have found applications in treating both endometriosis and adenomyosis [10,11]. This fact is based on the common features of the three pathologies at the molecular and genetic level [10,11,12,13]. The aim of this review is to identify those compounds from the relevant literature.

There appears to be a potential shortfall in funding for research on adenomyosis and endometriosis, possibly due to the perception that these conditions are non-fatal. However, this perspective needs reevaluation given the substantial social burden associated with patient management. Furthermore, while advancements in genomics, proteomics, metabolomics, and transcriptomics are anticipated, their outcomes have yet to meet expectations. A call for transparency in research practices is also essential [10].

## 2. Materials and Methods

The search employed specific keywords, including “endometriosis”, “adenomyosis”, “pathogenesis”, “endometrial cancer”, “malignant behavior”, and “personalized treatment”. A comprehensive literature review spanning the past decade was conducted using electronic databases, such as PubMed and Google Scholar. Initially, the search generated 4350 titles. Abstracts were meticulously examined based on inclusion criteria, excluding books, editorials, literature reports, and studies not aligned with this review’s objectives. The focus remained on peer-reviewed, full-text articles written in English that demonstrated proper study design, clarity, and informativeness. Duplicate findings, case reports, and studies with inappropriate designs or overlapping data were subsequently removed. Ultimately, 87 articles were chosen, and a narrative approach was adopted due to the unfeasibility of data pooling. Any discrepancies were addressed, and selected articles underwent rigorous evaluation for quality, reliability, and validity. The selection mode is presented in the flow-diagram (Figure 1).

## 3. Results

We emphasize here that the treatment aim for endometriosis and adenomyosis is different from endometrial cancer, where the outcomes have severe health consequences. The severe prognosis of endometrial cancer is given by the high recurrence rate and high mortality. The therapeutic approach for endometrial cancer is totally different from that for endometriosis and adenomyosis and consists of surgery and adjuvant chemotherapy and radiotherapy, which undermines the personalized treatment that is relevant in fertility preservation, recurrences, metastases or lack of response to classical therapy. The severity of cancer justifies the efforts of researchers to find a curative treatment for cancer and due to the complexity of pathogenesis and some common features with endometriosis or adenomyosis, these personalized treatments could seek applicability in endometriosis or adenomyosis as well. We present here the results regarding these various therapeutic strategies.

We have structurally divided the search results into two entities relevant to our review: pathogenesis and personalized targeted treatments based on pathogenesis. Before this, we emphasize some aspects related to endometrial cancer in which molecular classification is the basis of risk stratification in the therapeutic approach and the search for molecular-targeted therapy, with much more funding being allocated to this research compared to that on endometriosis and adenomyosis, but in which applicability is also found.

The EC risk stratification used in the treatment approach has been based on histological types. In 1983, Bokhman was the first to distinguish two types of EC [6]. Stage assignment is generally more objective, but trials have shown large inconsistencies and disagreements in grade, histology type, stage, or risk factors based on stromal invasion, myometrial invasion, or lymphovascular invasion (LVI). This leads to variations in clinical practice, which is associated with worse overall survival. This emphasizes the need for an objective EC classification system and also a common therapeutic management [8].

The gold standard therapy in endometrial cancer is surgery with or without adjuvant radiotherapy and chemotherapy. But in advanced cancer the options are limited, and histopathology and pathogenic mechanisms play the key role. According to the ESGO (European Society of Gynecological Oncology), ESMO (European Society of Medical Oncology), ESTRO (European Society for Radiotherapy and Oncology), and ESP (European Society of Pathology), molecular classification should be considered for EC [8]. It is also recommended to include this classification in standard pathology reporting and treatment decisions. The molecular classification based on mutations in DNA polymerase epsilon (POLE) and immunohistochemistry (IHC) yields four molecular subtypes: POLEmut, mismatch repair deficient (MMRd), p53abn, and NSMP (no specific molecular profile with normal p53 expression). This classification is highly reproducible and has an important prognostic value. POLE-mutated tumors are frequently of high-grade histology and their prognosis is significantly better than that in the other three groups. MMRd have an intermediate prognosis; p53 is the most aggressive; NSMT is the most common EC and has an intermediate prognosis [7,8,9].

This review highlights the need for personalized treatment based on similarities between endometriosis/adenomyosis and endometrial cancer based on their pathogeneses.

### 3.1. Pathogenesis

In terms of the etiopathogenesis and manifestation of the disease, common features of endometriosis and adenomyosis are similar to cancers, including the ability to evade apoptosis; stem cell-like dysregulation, migration, proliferation, and neovascularization; immune system alteration; an unusual formation of cytokines, growth factors, angiogenic factors; and changes in the expression of specific genes [1,11,12]. The diagnosis is established through histology.

In addition, there are established connections between endometriosis and certain types of cancers. The eutopic endometrium is the place at which primary defects in endometriosis can be located. Abnormalities might involve ectopic growth outside the uterine cavity. Evidence for the relationship between cancer and endometriosis is substantial: changes in the expression of tumor suppressor genes and oncogenes occurring in the eutopic endometrium lead to endometrial foci outside the uterus [1,11,13].

Among the etiopathogenesis theories for endometriosis, one that is widely accepted is retrograde menstruation, first described by Sampson in 1927. Several other theories have been developed over time, such as coelomic metaplasia, embryonic cell rest theory, and stem cell theory [1,14,15].

Endometriosis shares similarities with cancers in its genetic profile, characterized by mutations in tumor-suppressor genes such as PTEN (phosphatase and tensin homolog), ARID1A (AT-Rich Interaction Domain 1A), Tp53, and CTNNB1 (beta-catenin gene). While the precise mechanisms remain elusive, mutations in PTEN appear to influence cell cycle regulation, CTNNB1 mutations are associated with increased mobility and invasiveness, and mutations in PIK3CA (encoding the p110a catalytic subunit of PIK3) inhibit apoptosis. Furthermore, additional research has identified mutations in KRAS (Ki-ras 2 Kirsten rat sarcoma viral oncogene homolog), LOH (loss of heterozygosity), and BRAF (B-Raf murine sarcoma viral oncogene homolog B) within endometriotic lesions [1,3,6,11,16]. Additionally, endometriosis is recognized as a multifactorial disease, with evidence suggesting significant dysregulation of micro-RNA expression, mirroring its crucial role in cancer development [1,11]. Specifically, the downregulation of Mir-10b has been observed in both endometriosis and adenomyosis, indicating a shared molecular pattern [17].

Repeated hemorrhage in endometriotic lesions can contribute to carcinogenesis via increased oxidative stress, promoting DNA methylation and thus resulting in the activation of the anti-apoptotic pathway, which can alter the expression of genes. The aberrant methylation of DNA has been linked to endometrial pathologies, such as endometriosis, adenomyosis, and endometrial cancer, and could be a factor implicated in the hormonal instability observed in endometriosis as well as endometrial cancer [1,3,11,18]. This is a widely studied epigenetic process associated with the repression of gene activity. Epigenetic and genetic theories are currently being investigated to identify potential therapeutic targets [1,3,6].

Studies have shown that stem cells play an important role in the pathogenesis of endometriosis, adenomyosis, and EC. Stem cells are involved in the regenerative ability of the endometrial cycle [1,19]. Various studies have evaluated endometrial stem cells and their role in endometrial biology and proliferative conditions, such as endometriosis, adenomyosis, and endometrial cancer [20,21]. Evidence of endometrial regeneration via bone marrow-derived stem cells in patients receiving bone marrow transplantation suggests a new potential treatment. This is based on a new hypothesis for the etiopathogenesis of endometriosis. Extensive research is needed in stem cell biology to offer new opportunities for the diagnosis and treatment of endometriosis and cancers [1]. We will discuss their role in the setting of regenerative medicine as well as the treatment of these diseases [22].

In the early 1970s, Folkman highlighted the significance of angiogenesis in facilitating the growth and survival of tumor cells, suggesting it as a potential therapeutic target for malignancies by focusing on the endothelium. Interestingly, angiogenesis predominantly occurs in the endometrium, representing the sole healthy tissue where this process takes place [11]. Within the endometrial cycle, angiogenesis is more pronounced in the fundal region compared to the isthmic region of the uterus, leading to the substantial development of microvessels derived from circulating endothelial progenitor cells. In the context of endometriosis, there is an elevated density of microvessels along with increased expression of VEGF-A in the glandular epithelium and VEGF-2 in the endometrial blood vessels. Hormonal imbalances can potentially disrupt endometrial function [1]. While anti-angiogenic drugs are primarily recognized as targeted treatments for cancer, their therapeutic application has expanded to other conditions, such as endometriosis, cardiovascular diseases, and obesity [23]. Further research is essential to explore the relationship between altered gene expression, progesterone resistance, and angiogenesis [1].

The breakdown of natural immunological defenses, encompassing both cell-mediated and humoral immunity, can foster immune tolerance toward implanted tissue within the peritoneal environment. This phenomenon holds significant implications for both endometriosis and cancer. In endometriosis, a diminished activity of cytotoxic T cells and autoantibodies, coupled with a reduction in NK (natural killer) cells in the bloodstream, elevated macrophage levels, T cell depletion, iron-induced oxidative stress, inflammation, and hyperestrogenism, collectively establish a crucial link between endometriosis and cancer progression [1,3,11,18]. Furthermore, growth factors activated and secreted via immune endometrial cells stimulate the implantation and proliferation of ectopic endometrial tissue, along with promoting angiogenesis. The involvement of COX-2 also modulates the invasion of ectopic mesothelial cells, potentially paving the way for innovative treatment strategies in the future [1].

NK cells, integral components of the innate immune system within the uterus, play pivotal roles in menstruation, embryonic development, combating infections, and influencing cancer development. Additionally, the abundant presence of PD-L1 (programmed death ligand 1) in tumors correlates with poor prognosis [24]. Uterine NK cells, regulated via sex hormones, offer promising avenues for targeted therapies against uterine cancer if we gain a deeper understanding of their functions [25].

The immune system inherently regulates cancer pathogenesis by eliminating endogenous dead cells and safeguarding against alterations in the microenvironment that are conducive to tumor growth, thus maintaining homeostasis. Immuno-oncology focuses on deciphering the intricacies of the tumor immune microenvironment [25]. Key factors such as overexpressed adhesion molecules, matrix metalloproteinases, plasminogen activators, and iron overload, in addition to the involvement of macrophages and mast cells, contribute significantly to these processes [24,26].

Ultimately, preventive strategies emphasize reducing oxidative stress and modulating the peritoneal microbiome to mitigate the risk associated with these conditions [27].

The concept that endometriosis could also be considered an autoimmune disease is supported by the presence of autoantibodies, and this association with autoimmune diseases may inform future treatments. Immune cells could serve as potential therapeutic targets for both endometriosis and cancer [1,26,27].

Inflammation serves as another common pathogenic factor in the development of endometriosis/adenomyosis and cancer [11,28,29]. Endometrial microbiota and their association with inflammatory cytokines, including IL-6, IL-8, IL-17, and mRNA, play a role in endometrial cancer. Notably, a study by Wanting Lu (2020) found that only IL-6 levels were significantly elevated, and that Micrococcus showed a positive correlation with IL-6 and IL-17 mRNA levels, suggesting potential implications for future treatments [28].

Understanding the connection between microbiota and endometrial cancer is essential for both prevention and the development of innovative therapies [30]. Other studies have indicated that various factors, such as IL-1, IL-6, IL-8, TNF-alpha, IL-33, growth factors such as VEGF, platelet activation, immune cell activation (including macrophages, mast cells, T and B cells, along with reduced NK cell cytotoxicity), and stromal fibroblasts, collectively contribute to the formation of endometriotic lesions. Furthermore, oxidative stress, hypoxia, and the resultant iron release and overload are pivotal factors in this context [31].

Various other pathogenic mechanisms have been studied with the intention of targeting them in future treatments. For instance, Sphingosine 1-phosphate (S1P) is a bioactive sphingolipid that originates from sphingomyelin catabolism. It plays a role in the growth of various cells, including those involved in endometriosis or adenomyosis [32].

### 3.2. Personalized Treatment for Endometriosis, Adenomyosis/Personalized Treatment studied for Endometrial Cancers with applicability in Endometriosis or Adenomyosis

Molecular differences between eutopic and ectopic endometria pose challenges to developing new therapeutic strategies for these diseases. While surgery remains the gold standard for definitive diagnosis and treatment, there is a need for safer, more effective, and affordable therapies [33,34].

In the management of endometrial cancer, adjuvant chemotherapy is considered appropriate according to the guidelines of the ESGO (European Society of Gynecological Oncology), ESMO (European Society of Medical Oncology), ESTRO (European Society for Radiotherapy and Oncology), and ESP (European Society of Pathology) [8]. The purpose of this review is not to describe the standard and effective treatment in endometrial cancer, but the personalized treatment based on pathogenic mechanisms and molecular biology, which represents only a niche in treatment of this disease. However, personalized treatments targeting specific pathogenic mechanisms represent the future of treatment, particularly for addressing relapses [8]. Additionally, no cure currently exists for endometriosis and adenomyosis, which significantly impact quality of life and impose a societal burden [33,34].

In this review, we present management options for these pathologies, some of which are still under various stages of study. These diseases share common pathogenic mechanisms and we analyze here different compounds used across these pathologies.

Estroprogestins are commonly prescribed for managing pain associated with endometriosis and adenomyosis, as well as for menstrual cycle regulation. However, studies have not consistently demonstrated significant pain reduction. Their mechanism of action involves inhibiting ovulation, inducing decidualization, and causing atrophy. Common side effects include bleeding, breast pain, and headaches [4,10,33,35,36,37]. Estroprogestins are not recommended for treating endometrial cancers [38].

Progestins inhibit ovarian steroidogenesis, leading to hypoestrogenism, which promotes decidualization and exhibits anti-inflammatory effects, angiogenesis inhibition, and the suppression of epithelial-to-mesenchymal transition (EMT). They are particularly effective in treating pain related to endometriosis and adenomyosis and have a favorable long-term tolerability profile. Common side effects include breast tenderness, weight gain, fluid retention, bleeding, acne, headaches, mood swings, and elevated liver enzyme levels. Progestins commonly used for treating endometriosis and adenomyosis include etonogestrel-releasing implants, levonorgestrel-releasing IUDs, and Dienogest [4,8,10,11,33,35,36,37,39,40].

LNG-IUDs exhibit anti-inflammatory effects, induce endometrial atrophy, and exert a direct local action on adenomyosis and endometriosis, reducing uterine volume and associated pain. They can also decrease the need for surgical interventions [2,4]. Additionally, they may be employed in endometrial cancer management for fertility-sparing purposes, either alone or in combination with other medications, with a pregnancy rate of 75% [38].

Dienogest is effective in preventing the recurrence of lesions and pain following surgery, and it also improves quality of life. Experts recommend long-term treatment with Dienogest 2 mg > 15 months, with significant improvements observed in physical, mental, social, and general health compared with the baseline. This treatment is recommended post-surgery to prevent recurrence and can serve as an alternative therapy for adenomyosis; however, guidelines initially recommend the use of a levonorgestrel-releasing intrauterine device [4,10,33,36,37,39].

Additionally, Dienogest is utilized in endometrial cancer treatment. Studies have explored its efficacy in managing advanced, atypical endometrial hyperplasia and endometrial cancer. Dienogest influences cell proliferation, apoptosis, migration, and infiltration, suppressing cancer-derived cell lines, making it a viable option for clinical application to endometrial cancer. Experts have reported a successful response rate of 62/79 with a recurrence rate of 24% [41]. As a fourth-generation progestin, Dienogest can be employed in endometrial cancer cases where other progestins have proven ineffective. Progestins are also used in fertility preservation in EC before definitive surgery [42].

Gonadotropin-releasing hormone analogs are considered second-line therapies. These drugs, including Goserelin, Nafarelin, Leuprolide, and Triptorelin, suppress the production and release of gonadotropins, thereby inhibiting ovarian estrogen production. Common side effects associated with these drugs include depression, hot flashes, headaches, vaginal atrophy, decreased bone density, and alterations in lipid profiles.

Research has demonstrated a significant reduction in pain symptoms and an improvement in quality of life for individuals with endometriosis and adenomyosis when using GnRH analogs. However, a Cochrane review found no significant difference in efficacy between GnRH analogs and Danazol [2,4,10,11,33,35,36,37,40,43]. In the context of cancer treatment, these analogs inhibit proliferation and are involved in apoptosis [44].

Non-steroidal anti-inflammatory drugs (NSAIDs) are commonly used to alleviate pain symptoms associated with endometriosis and adenomyosis. However, long-term use of these drugs may lead to gastrointestinal ulcers, hypertension, and renal failure [2,4,35,36,37,40].

Metformin and heparin can also be utilized in the treatment of both endometriosis and endometrial cancer (EC) due to their anti-inflammatory effects. Metformin, traditionally known as an insulin sensitizer, has recently been identified as an anti-cancer agent. Insulin resistance is believed to play a role in the pathogenesis of EC, with elevated insulin levels being associated with both EC and lymph node metastasis, indicating a poorer prognosis [10,45,46]. An efficacy rate of 36.9% on the growth effect on epithelial cells and a 50% downregulation effect on Wnt expression were reported for metformin [45], and a 55.7% reduction in collagen gel contraction was reported for heparin [45]. Additionally, Nan Mu et al.’s study (2020) showed a 2.3-fold increased risk of mortality in their control group compared with patients receiving metformin [46].

Anti-TNF drugs (infliximab and etanercept) have the potential to inhibit inflammation; however, there are insufficient data to support their use specifically for endometriosis [47,48].

Additionally, various drugs are being explored as therapeutic options for pelvic pain and improving quality of life. These include Botulinum toxin, gepafixant, melatonin [80% efficacy rate], resveratrol [50% efficacy rate] [45], and quinagolide (reduce lesion size by up to 69.5%), niclosamide (reduce implant weight by up to 63.6%) [45]. Endorphin modulation is already utilized in managing chronic pain, autoimmune diseases, and cancers. Low doses of endorphin modulators stimulate endorphin release, while Naltrexone helps optimize compliance and alleviate symptoms, with a high rate of analgesia on supraspinal receptors and a lower rate on peripheral receptors [49,50]. Although these drugs show promising potential as multitarget therapies, further studies are needed to confirm their efficacy [45,49].

Gonadotropin-releasing hormone antagonists, such as Elagolix, may enhance long-term patient compliance. In the treatment of adenomyosis, Linzagolix has also been utilized with continued alleviation of symptoms, scoring 0 on various scales and improving quality of life, while the GnRH receptor antagonist Opigolix is currently under investigation. These antagonists do not induce the estrogen flare-up effect; instead, they promptly downregulate gonadotropin secretion by competing with endogenous GnRH for pituitary receptors and reducing steroid hormone levels. Similar to GnRH analogs, their effects are reversible, effectively alleviating pain symptoms. Common side effects include bleeding, headaches, nausea, anxiety, mild hot flashes, and lipid alterations [2,4,10,11,15,33,36,40,51,52].

The GnRH antagonist Cetrorelix exhibits antiproliferative effects on endometrial cancer cells but does not act through the GnRH type 1 receptor, with a study showing that cell count was reduced to 79.8 ± 3.9% of the control. It is considered a potential future treatment option for endometrial cancer [53].

Androgenic steroids, such as Danazol, induce the inhibition of pituitary gonadotropin-releasing hormones, suppress estrogen secretion, and act as inhibitors of local growth factors [33]. In a study by Claudia Tosti (2017), the cyclic versus continuous administration of vaginal Danazol were compared. The findings revealed better compliance with the cyclic administration of Danazol, with similar effects on reducing pain symptoms and improving quality of life [5]. Moreover, cyclic administration reduces androgenic side effects associated with vaginal use, such as hair loss, acne, hirsutism, hyperandrogenism, and weight gain [5,10,33].

The administration of Danazol may be a viable option for managing adenomyosis-related pain, particularly when utilizing a Danazol-loaded device [2,5,13,40,45,54]. Additionally, androgens can be employed to reduce the proliferation of endometrial cancer cells [55].

Aromatase inhibitors block estrogen production by inhibiting a crucial step in its synthesis from androgens. The third generation of aromatase inhibitors, primarily letrozole, has been successfully used in treating endometriosis and adenomyosis. Estrogen C18 steroids are characterized by the presence of an aromatic ring. While estrogen is predominantly produced in the ovaries, postmenopausal women generate it in extragonadal sites, with one key step being catalyzed via the aromatase enzyme. Many tissues express this enzyme.

Aromatase inhibitors are currently approved for clinical use in oncology and endometriosis treatment. They selectively inhibit estrogen production without affecting other steroidogenesis enzymes. Administered orally, these inhibitors exhibit rapid clearance from the body due to their short half-life, preventing accumulation in tissues. They are generally well tolerated, with mild side effects such as hot flashes, headaches, decreased bone density, weight gain, fatigue, depression, spotting, insomnia, and reduced libido. They prove more effective when used in conjunction with GnRH analogs and are typically not recommended as a first-line therapy. An efficacy rate of between 29 and 61% in decreasing CA-125 serum levels as well as a fertility rate of between 38 and 100% [56] have been reported, highlighting their significant drawbacks [10,13,33,35,36,40,56,57,58]. However, there are limited available data supporting their efficacy [2,13,40,56].

Anti-angiogenic treatment may be beneficial for early-stage diseases but requires chronic administration and could potentially prevent recurrence after surgery in conditions such as endometriosis, adenomyosis, or cancers [23]. Its mechanism leads to tumor ischemia and necrosis, exhibiting procoagulant and proapoptotic effects. However, there are potential drawbacks to anti-angiogenic therapy for endometriosis, such as the need for chronic administration and the inability to target mature vessels in endometriotic lesions. Combining anti-angiogenic agents with inhibitors of VEGF, growth factors, and platelet-derived growth factors may enhance their efficacy in treating deep endometriosis. Due to the heterogeneity in neovascularization, a single angiostatic compound may not suffice, but when combined with other hormonal therapies, it can be beneficial. While no obvious adverse effects have been observed, this treatment may not be suitable for fertility treatments as it could completely inhibit embryonic growth [59].

Anti-angiogenic drugs such as bevacizumab, an antibody against VEGF, can inhibit endometriotic lesions by inducing apoptosis. Similarly, tyrosine kinase inhibitors (TKIs) that target the activity of tyrosine kinase receptors, such as the anti-VEGF cediranib, are under investigation [7,36,40,45]. In an animal model, Sunitinib has been shown to cause a decrease of 78.8% in lesion size [45]. Dopamine and cabergoline play critical roles in regulating VEGF-mediated growth of endometriotic lesions. Other substances such as statins and caffeic acid can also suppress endometriotic and adenomyotic lesions [7,36,40,45].

In the context of endometrial cancer, anti-angiogenic treatment has shown promise when used in combination with chemotherapy and immunotherapy [23]. Although bromocriptine and cabergoline show promise in experimental models through modulating pro- and anti-angiogenic pathways, they have not yet been widely adopted. Pentoxifylline, an anti-plasminogen activator inhibitor-1, may increase pregnancy rates in women with endometriosis without interfering with ovulation [47].

Additionally, resveratrol—a phytoestrogen found in grapes, wine, soy, berries, and other dietary sources—possesses anti-inflammatory, anti-angiogenic, and anti-proliferative properties. Its mechanisms are based on inhibiting prostaglandin synthesis and inducing apoptosis, positioning it as an innovative drug for preventing and treating endometriosis [10,25]. In an animal model, it reduced lesion size by 41.5% [45].

Selective estrogen receptor modulators (SERMs) target estrogen receptors in cells. To date, there is no evidence supporting their beneficial effects on endometriotic or adenomyotic lesions, and they may even exacerbate hyperalgesia [10,35,36]. SR-16234, identified as a novel selective estrogen receptor modulator by Tasuku Harada in 2017 [60], exhibits estrogen antagonistic activity and has demonstrated strong inhibitory effects on transplanted endometrial tissue. In long-term administration, a statistically significant reduction in pelvic pain and cyst size was observed; however, this clinical trial involved only 10 patients without a control group [60]. According to J.V. Pinkerton in 2019, raloxifene, bazedoxifene, and other SERMs have shown limited efficacy against endometrial cancer [61].

Regarding selective progesterone receptor modulators, mifepristone has been clinically applied since 1982. It can act as either an agonist or antagonist at the progesterone receptor, leading to cell cycle arrest. Mifepristone induces cell apoptosis through the mitochondria-dependent signaling pathway in endometrial epithelial cells and stromal cells of adenomyosis. It also inhibits ovulation in both adenomyosis and endometriosis and suppresses the migration of endometrial epithelial and stromal cells by inhibiting the epithelial–mesenchymal transition (EMT) in adenomyosis. Additionally, it reduces uterine volume and exhibits antitumor effects on endometrial cancer. However, significant side effects include bleeding, pelvic pain, headaches, and nausea [2,5,10,33,36,40,62].

In a 2019 study by Xuan Che involving 20 patients with adenomyosis treated with mifepristone, it was found to induce G1/G0 and G2/M phase arrest in the cell cycle of endometrial epithelial cells [63]. According to Lukovic (2021), it induced a 3-fold increase in apoptosis in endometriotic cells compared to their control group [62]. Mifepristone also inhibits the migratory capacity of eutopic endometrial epithelial and stromal cells in adenomyosis, induces apoptosis in both eutopic and ectopic endometrial cells, and inhibits EMT [63,64].

According to Francesca Conway (2018), ulipristal acetate, also known as a selective progesterone receptor modulator (SPRM), exacerbated ultrasound features and painful symptoms in adenomyosis, even though it dramatically decreased CA-125 concentration in serum [65]. Currently, SPRMs are not recommended for the treatment of endometriosis or adenomyosis due to the numerous adverse effects, including headaches, mood changes, and liver failure. Ulipristal is associated with a significant risk of endometrial carcinoma and hepatic damage. Interestingly, while it exhibits an antiproliferative effect on endometrial cancer cells, it also triggers the activation of pro-inflammatory cytokines. The only notable improvement observed is a reduction in bleeding [2,10,33,65,66,67].

Selective androgen receptor modulators (SARMs) have rekindled interest in targeting receptors for endometrial disorders, such as endometriosis, adenomyosis, or endometrial cancer, due to their antiproliferative effects [10,55,68].

Antioxidants are crucial because oxidative stress plays a significant role in the production of cytokines, prostaglandins, and reactive oxygen species. Various antioxidants have been studied in animal models. Omega-3 helps improve post-surgery pain, while N-acetylcysteine can reduce the volume of endometriotic lesions. Statins significantly decrease proliferation in endometriotic tissue. Metformin exhibits anti-inflammatory effects and modulates estrogen production. Vitamins D and A reduce lesions and possess anti-adherent properties that reduce inflammation. Additionally, N-acetylcysteine and catalase reduce levels of reactive oxygen species (ROS) and exhibit a beneficial impact on autophagy in endometrial cancer [10,25,36,45,69]. Caffeic acid was found to reduce catalase by 56% [45]. Some studies have focused on plant-derived compounds from traditional Chinese medicine, exploring their activity to discover multitargeted drug molecules for rational therapy [4,45,70].

Immunotherapy may become increasingly important, given the successes seen in oncology. Natural killer (NK) cells play a role in the surveillance and apoptosis of cells within endometriotic lesions. Due to the limitations of current treatments, researchers are exploring new approaches [49]. However, immunomodulators have not lived up to their expectations, failing to reduce the size or number of endometriotic implants or alleviate pain. In this context, inhibitors of Interleukin-1 Receptor Associated Kinase 4 (IRAK-4) are under investigation [10,36,37]. TNF antagonists have shown effectiveness in animal models.

Vaccines using Bacillus Calmette–Guérin (BCG) are among the most effective immunotherapeutic agents that stimulate NK cells and prevent the typical IL6, IL10, and IL4 responses observed in endometriosis. In immune-protected rats, the use of BCG-based vaccines decreased the probability of endometriosis induction from 69.6% to only 4.3%. Modulating NK cells, which possess inhibitory receptors on their surface that can suppress their activity against malignant or ectopic endometrial cells, may also be beneficial in treating endometrial cancer. The PD1 receptor, which binds to the PD-L1 ligand, has been successfully implemented in cancer treatments [49].

Recent data highlight the impact of programmed death 1 and programmed death ligand 1 (PD1/PD-L1) inhibitors on chemo-resistant metastatic endometrial cancer. Drugs such as pembrolizumab and dostarlimab are currently under study, showing promising results. Additionally, the angiogenesis inhibitor lenvatinib is being tested, sometimes in combination with other immune inhibitors. These combination therapies are undergoing evaluation in randomized studies as encouraging strategies to combat immunotherapy resistance. According to Menhert (2016) [71], pembrolizumab may be used in cases of mutant endometrial cancer, serving as a monoclonal antibody treatment for patients with MMR tumors who have previously undergone chemotherapy, as supported by phase II studies [24,71,72].

According to a study by Xishuang Wang (2020), IL-37bdelta1-45 suppressed the migration and invasion of endometrial cancer cells compared to their control group (*p* < 0.05). Interleukin-37 is a recently discovered anti-inflammatory cytokine that belongs to the IL-1 family, considered the most biologically functional subtype. It plays a protective role against various types of cancers and also has significance in conditions such as endometriosis and adenomyosis [73].

Various new pathways are being explored for targeted treatment in oncology, particularly for cases of relapse or cancers that do not respond to conventional therapy. Targeting HER2/Neu has shown improved overall survival in a prospective randomized phase II clinical trial when trastuzumab was added to paclitaxel–carboplatin-based chemotherapy. The ERBB2 oncogene encodes HER2/Neu proteins and is associated with a negative prognostic indicator [74].

Efforts are also underway to target the PI3K-AKT-mTOR pathway. Both the phosphoinositide 3-kinase (PIK3) and mammalian target of rapamycin (mTOR) signaling pathways are being investigated. Alterations in these pathways in endometrial cancer (EC) lead to the angiogenesis and proliferation of aberrant cells. While mTOR inhibitors such as temsirolimus and ridaforolimus continue to be studied in relation to EC, their combination with other agents has shown significant toxicity [7,75]. Additionally, the use of mTor inhibitors in advanced or recurrent diseases has shown only modest activity (<25% response rates) [7]. However, the combination of everolimus with letrozole has demonstrated a favorable and durable response [7,75]. The PIK3/AKT/mTOR and AMPK-mTOR pathways, along with mechanisms of autophagy, are under investigation as potential therapeutic targets in clinical management [8,76].

The Wnt signaling pathway plays a significant role in proliferation, metastasis, and chemotherapy resistance. Both its role as a biomarker and its potential for targeted therapy development are under investigation. Its involvement in endometrial cancer (EC) is not yet fully understood. Recent findings have suggested a connection between Wnt signaling and estrogen and progesterone receptors, with emerging evidence pointing to the mTOR signaling pathway, as noted by Iram Fatima (2021) [16].

Niclosamide, a salicylamide derivative, targets the Wnt/β-catenin pathway. Salinomycin, an antibiotic, induces apoptosis and disrupts the Wnt/β-catenin pathway, significantly reducing proliferation, migration, and invasion in EC. Curcumin inhibits carcinogenesis and promotes apoptosis, along with other dietary products. Additionally, miRNA treatments may target at least 200 genes; their downregulation promotes the epithelial–mesenchymal transition (EMT) and is associated with Wnt signaling [16].

ARID1A (AT-rich interactive domain 1A) is the most frequently mutated gene among all chromatin remodeling genes and is detected in many malignancies, particularly in endometrial cancer (EC). It can serve as a biomarker for identifying the early stages of cancer. ARID1A mutation may be associated with cancer cell sensitivity to EZH2 inhibitor therapy as well as PARP inhibitor drugs due to its role in DNA damage repair. Its potential benefits are being evaluated in clinical trials [77,78].

Epigenetic agents are utilized as targeted treatments for endometriosis and adenomyosis, in particular to reduce cytokine production. According to V. Tskhay, it is essential to implement appropriate personalized treatment post-surgery, incorporating epigenetic therapy that influences the pivotal chain of pathological processes [35]. Valproic acid has been shown to effectively reduce the size of endometriotic lesions in mice [36]. Anti-platelet therapy holds promise for future treatments, given the significant role platelets play in the pathogenesis of adenomyosis and endometriosis. These lesions undergo repeated tissue injury and repair, with platelets contributing to epithelial–mesenchymal transition and fibrosis. Anti-platelet therapy may play a role in reducing pain and fibrosis [2,10].

Higher activities of sulfatase and sulfotransferase (STS) have been observed in cancerous endometrium, as well as in cases of endometriosis and adenomyosis. Clinical trials involving sulfatase inhibitors, such as Irosustat, are under discussion. However, clinical studies evaluating STS inhibitors in gynecological diseases have not yet yielded convincing data. Sulfatase inhibitors primarily target endometrial cancer (EC) [68,79].

Migration mechanisms can also be targeted. Tensin-1 is a protein found in deep endometriotic adhesions that may play a role in adhesion and migration processes. The expression of the TNS1 protein and mRNA was found to decrease in patients undergoing treatment with GnRH agonists [47]. Additionally, targeting dysperistalsis through oxytocin and vasopressin receptors using atosiban could be a valuable approach to managing adenomyosis. There are currently no studies published or registered on the use of agents targeting peristalsis via oxytocin or vasopressin receptors in the treatment of adenomyosis or EC [40].

Endometrial stem cells offer a promising therapeutic avenue to inhibit the growth of endometriotic lesions and enhance outcomes for endometrial cancer (EC) [19,47]. These stem cells can transform into tumor cells, contributing to conditions such as endometriosis, adenomyosis, and EC, which are often accompanied by genetic and epigenetic alterations. Lovastatin has shown potential in promoting the differentiation of endometrial stem cells while reducing the expression of stemness markers, with epigenetic factors playing a role, making statins a viable treatment option [21,22]. Additionally, selective estrogen receptor modulators, such as raloxifene and bazedoxifene, have demonstrated the ability to reduce bone marrow stem cell seeding at ectopic endometrial sites [10,22].

Furthermore, the genomic alterations observed in EC offer crucial insights into its pathogenesis, guiding future therapeutic strategies. To determine the most cost-effective approach for managing endometriosis, adenomyosis, and EC, there is a pressing need for large, well-designed randomized trials [2,13,40,56]. Therapeutic options are briefly presented in the following table (Table 1).

## 4. Discussions

In this review, we outline treatment options for these conditions based on their respective pathogeneses. Due to several common features, various therapeutic approaches have been explored across all three pathologies, yielding varied results. We also recognize the critical distinction in severity and health consequences between endometrial cancer and these benign diseases. The purpose of the manuscript is not to detail the standard treatment of EC for which there are already indisputable clinical guidelines, but to explore personalized treatment that targets various pathogenic pathways and may also be applicable in endometriosis or adenomyosis.

In developed countries, endometrial cancer (EC) is the most common malignancy of the female genital tract, typically affecting postmenopausal women. Recently, the ESMO, FIGO, and ACOG have recommended a systematic surgical staging approach, which is now considered the gold standard for initial treatment. Additionally, postoperative adjuvant treatments, including radiotherapy and chemotherapy, are crucial in preventing recurrence and metastasis. However, for advanced-stage cancers, traditional treatments have proven ineffective. Histology remains essential, and in this context, molecular classification is the current recommendation. A risk stratification approach is essential [80]. Molecular therapies targeting various signaling pathways (EGFR, VEGFR, and PI3K/PTEN/AKT/mTOR) have been developed over the past decade. When used post-surgery, they have achieved only modest response rates thus far, unless combined with chemotherapy or radiotherapy [80].

Endometriosis and adenomyosis impose a significant burden on the healthcare system. Due to the heterogeneity of symptoms, there’s often a delay in diagnosis, and available therapeutic options are not curative. Currently, there’s increased emphasis on understanding the pathogenesis of these diseases [1,32,60,66].

Despite numerous drugs being tested, only a few have been implemented in clinical practice [81]. NSAIDs (nonsteroidal anti-inflammatory drugs), oral contraceptives, and progestins primarily alleviate pain initially. However, long-term clinical evidence supporting their use is lacking, and they come with potential side effects. Nonetheless, their cost-effectiveness compared to other drug types is favorable. As for nonsteroidal anti-inflammatory analgesics, the options are limited. COX-2 inhibitors carry significant side effects but are considered first-line medications for treating endometriosis-associated pain. While medical treatments alleviate pelvic pain in conditions such as endometriosis and EC, the pain tends to recur with chronic treatment due to drug side effects [10,36,45,46,47,48,50,66,82].

GnRH agonists are considered to be the next option when first-line therapies prove ineffective, are not tolerated, or are contraindicated. They are not recommended for long-term use due to their cost and significant side effects, notably bone density loss. In advanced or metastatic endometrial cancers, GnRH analogs exhibit only a marginal effect [10,35,36,37,43,44,66].

Aromatase inhibitors are reserved for women who do not respond to other treatments. They are beneficial for treating deep endometriosis and endometrial cancer (EC). These inhibitors should be prescribed in combination with progestagens or GnRH analogs for increased effectiveness and fewer mild side effects. However, more randomized clinical trials are necessary to provide conclusive data [10,36,56,58,66].

Among newer drugs, gonadotropin-releasing antagonist medications show promising results. Elagolix, a next-generation GnRH antagonist, has demonstrated efficacy in managing endometriosis-associated pain and slowing disease progression. The most significant side effect is a decrease in bone mineral density, which varies depending on the dosage. Proellex, a selective progesterone receptor modulator, exhibits progesterone antagonist activity. A major concern with this drug is its potential to alter the eutopic endometrium and cause hepatic damage. Despite these concerns, it has shown effectiveness in inhibiting cell proliferation in EC and warrants consideration for treatment [2,4,10,11,15,33,36,37,40,43,51,52,53,65,66].

In their 2018 study, Takashi Matsushima evaluated a combination of GnRH agonists with low-dose estrogen and progestin, specifically LEP/Dienogest (DNG), for 16 weeks. They observed that GnRH demonstrated efficacy in reducing uterine volume, while the other two treatments did not prove effective [83]. Further studies are required, as potential future therapies are currently being tested only in animal models.

Traditional treatments, such as gestrinone or Danazol, should be reserved for cases where other treatments have failed and when side effects are absent, as only progestagens have demonstrated some efficacy [38,41,42,66].

Ulipristal is linked to a heightened risk of endometrial carcinoma and liver damage, with its primary benefit being a reduction in bleeding [66,67].

The use of Chinese herbal medicine has not demonstrated clear evidence of efficacy. Cannabis provides only moderate pain relief [66]. However, according to Xin Wang, Chinese herbal medicine promotes blood circulation and removes blood stasis. These herbs exhibit effectiveness against inflammatory cytokines, such as IL-6, IL-8, and TNF-alpha; inhibit cancer growth; regulate VEGF expression; and modulate the immune system. They also possess anticoagulant and anti-platelet properties. Given these potential benefits, Chinese herbal medicines warrant further exploration. There is a pressing need for global attention and comprehensive studies on this subject [84].

Regarding lifestyle, diet, and exercise, there are insufficient comprehensive studies to provide compelling evidence that could improve quality of life [66].

In terms of management, additional research is essential. This includes personalized treatment approaches, anti-angiogenic factors, immune-modulating drugs, and empirically addressing pain with appropriate analgesics, as well as contraceptive drugs or progestagens [4,8,10,11,24,33,35,36,37,40,49,59,66]. Furthermore, studies are required to understand the role of endometrial mesenchymal stem cells in adenomyosis and endometrial cancer. Targeted treatment strategies are crucial, and the integration of stem cells with biomaterials appears to be a promising therapeutic avenue for the future [19,21,22].

Endometriosis and adenomyosis may be associated with cancers, typically presenting a more favorable prognosis for malignancy. According to Zhengmao Zhang’s (2018) study, the presence of adenomyosis in endometrial cancer patients, especially those with a postmenopausal status, is linked to a lower grade of cancer and improved outcomes. This suggests that adenomyosis serves as a protective factor leading to better patient outcomes [85]. However, Hulya Ayik Aydin’s (2017) research indicates that adenomyosis does not significantly impact survival rates or patient outcomes and does not influence cancer treatment decisions [86].

Regrettably, we currently lack curative treatments for these diseases, particularly in cases that are advanced or recurrent. There is ongoing discussion about new molecules that target hormonal pathways. Researchers are exploring additional biological targets related to estrogen and progesterone to address the limitations of long-term medication. Studies are being conducted on drugs that affect estrogen synthesis and metabolism, address progesterone imbalance, evaluate nuclear progesterone receptor responsiveness, and explore epigenetic alterations of hormonal responses using demethylation or histone deacetylase inhibitors like valproic acid [87].

The disappointing outcomes or failures of trials, coupled with a lack of transparency and concerning trends in drug innovation for endometriosis, adenomyosis, and endometrial cancers, have contributed to inconclusive results. For instance, the raloxifene trial indicated exacerbating effects, the Infliximab trial involving an anti-TNF alpha antibody showed inefficacy, trials on selective progesterone receptor modulators (SPRMs) revealed serious side effects, and trials on oxytocin receptor antagonists have not yielded beneficial results [10]. The efficacy of bevacizumab, a well-described VEGF inhibitor antibody, has been extensively evaluated for treating endometrial cancer (EC) over the past 12 years; yet, only a few responses have been observed. When added to standard carboplatin–paclitaxel therapy in a first-line setting, bevacizumab did not provide any survival benefits compared to chemotherapy alone [24].

The development of cancer is linked to chronic inflammation due to interferon production. Although PARP inhibitors were explored in early-phase studies presented at the 2020 ESMO meeting, only partial responses were noted, with hematologic disorders identified as side effects. Nevertheless, understanding immune responses and modifying the immune microenvironment remains crucial for EC, despite the challenges in identifying responders. Further research is necessary to elucidate the resistance mechanisms to immunotherapy [24].

The high risk of recurrence, coupled with long-term medical therapies, often leads to intolerable side effects and poor compliance [29].

Hysterectomy, with or without adjuvant therapy, remains the gold standard for treating endometrial cancers. However, therapeutic options become limited in advanced stages. The recommended adjuvant therapy involves a taxane–platinum combination; yet, no standard second-line therapy exists. Since molecular classification fails to elucidate varying therapy responses, the focus has shifted to the tumor immune microenvironment, which may pave the way for new immuno-oncology targeted therapies [24]. While chemotherapy remains the primary treatment approach for most patients, advancements in immunotherapy are shaping new management strategies. Ongoing trials aim to maximize the benefits of immune checkpoint inhibitors for patients with EC [7].

Efforts are made for the diagnosis of endometriosis and adenomyosis, increasing the accuracy of imaging criteria along with the progress achieved in pathogenesis [88,89]. In the case of targeted treatments, new research approaches would be useful.

There is a pressing need for greater transparency regarding clinical trials. Often, the reasons for the failure of these trials remain entirely unclear, and there is a lack of open discussion. Without addressing the mistakes from these unsuccessful trials, there is a risk of repeating them in future research endeavors. For these pathologies, particularly in advanced stages, no definitive treatment can be recommended. The absence of high-quality prospective studies significantly contributes to this predicament [13]. Various new pathways are being researched (Table 2).

## 5. Conclusions

An unmet need exists for the cure of the studied diseases. Long-term treatment becomes necessary because symptoms persist, and surgeries are often followed by postoperative recurrence. We emphasize the necessity for new, effective, long-term treatments based on pathogenesis while considering their potential adverse effects. From this standpoint, particular attention should be given to understanding the pathogenic mechanisms shared by these diseases. Such an understanding could pave the way for personalized treatment strategies.

## Figures and Tables

**Figure 1 pharmaceuticals-17-00311-f001:**
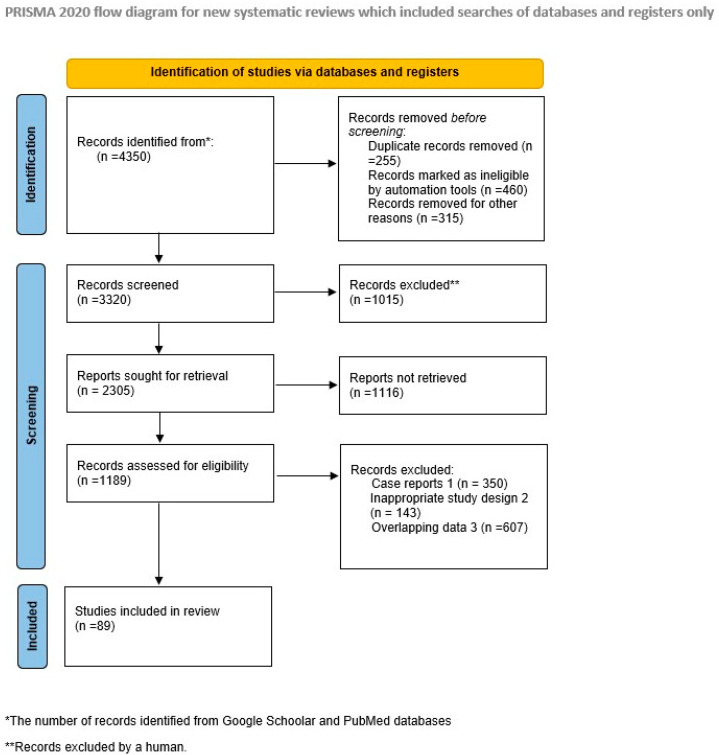
PRISMA flow chart. Identification of studies via databases and registers.

**Table 1 pharmaceuticals-17-00311-t001:** Therapeutic options for endometriosis, adenomyosis, and endometrial cancer based on pathogenesis.

Therapeutic Options	Pathologies	Symptoms	Mechanism	Side Effects
Estroprogestins [4,10,33,35,36,37]	Endometriosis Adenomyosis	Pain Control of menstrual cycle	Ovulation inhibitor Decidualization Endometrial atrophy	Vaginal bleeding Breast pain Headaches
Progestins [4,8,10,11,33,35,36,37,40,41,42,70]	Endometriosis Adenomyosis Endometrial cancer	Pain Decreased uterine volume Reduced need for surgery Fertility-sparing management Lesion recurrence prevention Improved quality of life	Suppress ovarian steroidogenesis Decidualization Anti-inflammatory effects Anti-angiogenesis Reduced cell proliferation, migration, apoptosis, and infiltration in EC	Breast pain Weight gain Fluid retention Bleeding Acne Headaches Mood changes Liver enzyme increase
GnRh analogs [2,4,10,11,33,35,36,37,40,43,44]	Endometriosis Adenomyosis Endometrial cancer	Pain Preventing the recurrence of pain and lesions	Suppress ovarian estrogen production Inhibit cell proliferation and migration	Depression Hot flashes Headaches Vaginal atrophy Bone density decrease Alteration of lipid profile Induce apoptosis
Anti-inflammatory drugs: NSAIDs [2,4,35,36,37,40] Metformin, Heparin [10,45,46] Anti-TNF [47,48] Botulinum toxin Melatonin [45] Naltrexone [49]	Endometriosis Adenomyosis Endometrial cancer	Pain	Under study for their anti-inflammatory effects (insufficient data)	Gastro-intestinal ulcers Hypertension Renal failure
GnRh antagonists [2,4,10,11,13,15,33,36,40,51,52,53]	Endometriosis Adenomyosis Endometrial cancer	Pain	Some are under study Downregulation of estrogen Anti-proliferative effects	Vaginal bleeding Nausea Headaches Anxiety Mild hot flashes Lipid modification
Androgenic steroids (Danazol) [2,5,10,13,33,40,45,54,55]	Endometriosis Adenomyosis Endometrial cancer	Pain Good compliance	Inhibition of pituitary gonadotropin-releasing hormone Inhibition of estrogen secretion Local growth factor inhibitor Decrease in proliferation	Hair loss Acne Hirsutism Weight gain Vaginal bleeding
Aromatase inhibitor [2,10,33,35,36,40,56,57,58]	Endometriosis Endometrial cancer	Pain	Blocks estrogen synthesis from androgens	Hot flashes Headaches Bone density decrease Weight gain Fatigue Depression Insomnia Spotting Decreased libido
Anti-angiogenic treatment [10,23,25,36,40,45,47,59]	Endometriosis Adenomyosis Endometrial cancer	Beneficial in early-stage disease Prevents recurrence	Ischemia and tumor necrosis Procoagulant Proapoptotic effect Antibody delivers toxic agents against tumor endothelium	No obvious adverse effects
SERMs [10,35,36,61]	Endometriosis Adenomyosis Endometrial cancer	Various beneficial effects Pain Reduce lesion size	Target estrogen receptors	Enhance hyperalgesia Sometimes exhibits estrogenic effects
SPRMs [2,5,10,33,36,40,62,63,64,65,66,67]	Endometriosis Adenomyosis Endometrial cancer (under study)	Pain Decrease in uterine volume Antitumoral effects	Cell apoptosis Inhibition of ovulation Suppress EMT Inhibits migration	Can induce endometrial cancer Liver failure Mood change Headaches Pelvic pain Nausea Vaginal bleeding
SARMs [10,55,68]	Endometriosis Adenomyosis Endometrial cancer (under study)	Pain Muscle loss Urinary stress incontinence	Anti-proliferative effects Modulate androgen-like steroid receptors	Unknown
Antioxidants [4,10,25,36,45,69,70]	Endometriosis Adenomyosis Endometrial cancer (under study)	Pain Reduce lesions	Anti-inflammatory effects Anti-proliferative Anti-adherent Anti-angiogenic Affect apoptosis and autophagy	Still under investigation
Immunotherapy [10,24,36,37,49,71,72]	Endometriosis Adenomyosis Endometrial cancer (under study)	Reduces the size and number of lesions, but controversial results	Anti-inflammatory effects Reduces ROS	Still under investigation
Epigenetic agents (valproic acid and anti-platelet agents) [2,10,35,36,40]	Endometriosis Adenomyosis Endometrial cancer (under study)	Reduce lesion size Reduce pain and fibrosis	Anti-inflammatory effects Effects on EMT	Unknown
Stem cells [10,19,21,22,47]	Endometriosis Adenomyosis Endometrial cancer (under study)	Pain Reduce lesion sizes	Induce cell differentiation Reduce stemness	Unknown

**Table 2 pharmaceuticals-17-00311-t002:** Various new pathways for targeted treatment.

Therapeutic Research Targeting New Pathways	Pathologies	Outcome	Disadvantage
HER2/Neu [Natalia Buza et al., 2014] [74]	Endometrial cancer	Improved free survival	Must be combined with classical chemotherapy
Targeting PI3K-AKT-mTor pathway (temsirolimus, ridaforolimus, and everolimus; at least five clinical trials are being conducted) [7,75]	Endometrial cancer	Only everolimus showed a good response	High toxicity
ARID1A and EZH2 inhibitors [77,78]	Endometrial cancer	Disappointing efficacy as monotherapy	Must be combined with other drugs
Sulfatase and sulfotransferase [68,79]	Endometriosis Endometrial cancer	No convincing clinical outcomes	Limited effects
PARP, PD-1, and PD-L1 [24,49,74,78]	Endometriosis Endometrial cancer	Partial response	PARP inhibitors associated with hematologic disorders

## Data Availability

Data sharing is not applicable.

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
