# Peer review of "Shared Pathogenic and Therapeutic Characteristics of Endometriosis, Adenomyosis, and Endometrial Cancer: A Comprehensive Literature Review"

_pharmaceuticals, 2024, doi:10.3390/ph17030311_

Round 1

Reviewer 1 Report

Comments and Suggestions for Authors

This is a review of therapeutic options for endometriosis, adenomyosis and endometrial cancer. Is is a comprehensive review and the methodology of the literature search is adequately presented. The manuscript should be shortened. In the introduction section the authors should provide justification why these three pathologies are included in a single review? What is the malignant potential of endometriosis and adenomyosis? Please omit the Type I and II of EC which is obsolete and not used any more. What is the prognosis of all 4 groups of EC according to molecular classification? Finally, when writing about promising targeted therapies in EC the response rates for various therapies should be included in the text

Author Response

Reviewer 1  

Dear reviewer

We tried to shorten it to be less boring and this manuscript represent 65% of the initial article.

We pointed out in the introduction the reason why these three pathologies are included in a single review, adding relevant material: line 44-50, 52-56,94-100.

Malignant transformation is rare and in general the prognosis is good. But the exact mechanism of those transformations are still unknown. [line 55-56].

We omit, as you asked, the description of type I and II of endometrial cancer. [line 76] We added the prognosis of all 4 group of EC according to the molecular classification: POLE-mutated tumors are frequently of high-grade histology, their prognosis is significantly better than that in the other three groups. MMRd have an intermediate prognosis, , p53 is the most aggressive, NSMT is the most common EC and has an intermediate prognosis. [7,8,9].

Related to the rate of response we added where available : In the treatment of adenomyosis, Linzagolix has also been utilized with continued alleviation of symptoms scoring 0 on the scales and a very good quality of life [line 353-355].

Related to estroprogestins: . However, studies have not consistently demonstrated significant pain reduction.[line 293-294]

Related to LNG-IUD in early stage EC the authors reported a pregnancy rate of 75% [38].[line311]

 Experts recommendations consists in long-term treatment with dienogest 2mg > 15 month with signifficant improvements  in physical, mental, social, general health compared with baseline. [line 314-316].

An efficacy rate of 36.9% on growth effect on epithelial cel land 50% downregulation effect on wnt expression was reported for metformin[45] and a 55.7% on collagen gel contractin was reported for heparin [45].

Also Nan Mu et all (2020) have shown a 2.3-foldincreased risk of mortality in the control group compared with patients receiving metformin.[46]

The GnRH antagonist Cetrorelix exhibits antiproliferative effects on endometrial cancer cells but does not act through the GnRH type 1 receptor, the cell number was reduced to 79.8+_ 3.9% of control. It is considered a potential future treatment option for endometrial cancer [53].

Other rates or various failures I emphasized in the text.

Thank you very much for your suggestions.

Reviewer 2 Report

Comments and Suggestions for Authors

In line 69 ‘Over 9% of EC occurs in women above the age of 50.this sentence is not correct.

In line 85 ‘The gold standard therapy in endometrial cancer is surgery with or without adjuvant chemotherapy’. Should be adjuvant radiotherapy and chemotherapy

In line  97-98 ‘The aim of this review is to identify compounds from relevant literature utilized in three pathologies known for their malignant behavior: endometriosis, adenomyosis, and endometrial cancer’. Endometrial cancer is a malignant pathology, but others are benign pathologies. So comparison of malignant and benign pathologies for their malignant behavior is not seen reasonable. It may be necessary to subtract endometrial cancer from this article. 

In line 174  ‘Tp53’ should be removed.  There is p53 in same sentence.

There is close relationship between endometriosis and some cancer such as ovarian endometriod and clear-cell cancer. But there is no any acceptable relation among endometriosis and uterine endometrial cancer. It is better to remove subject of endometrial cancer from this article, although there may be some molecular similarity and also some low degree therapeutic similarity among endometriosis,  adenomyosis and endometrial cancer.

Author Response

Reviewer 2

Dear reviewer:

Yes, I corrected the percentage, it is 90% [Line 69].

Line 85 I corrected as you suggested.

Line 97-98 This observation related to malignant behaviour of endometriosis and adenomyosis was studied in several manuscripts. It is not something new. And the special issue in which I chose to publish is about endometriosis, adenomyosis and endometrial cancer, probably because they have common features. Endometriosis and adenomyosis can undergo malignant transformations. And also, as I mentioned in the manuscript, medications initially developed for endometrial cancer have found applications in treating both endometriosis and adenomyosis [line98-99]. As presented, we are talking about personalized targeted treatment, not including surgery, radiotherapy and chemotherapy.

While Tp53 is the gene that codes the protein p53,  I removed p53 and I kept Tp53 because the whole phrase is about genetics. Thank you for your observation because I missed it.[line 173]

You are right that there is no established link between endometriosis and endometrial cancer except for some molecular similarities even if some researcher fiand the similarityes substantial:  Evidence for relationshipe between cancer and endometriosis are substantial: changes in the expression of tumor suppressor genes and oncogenes occuring in the eutopic endometrium leads to endometrial foci outside the uterus [1,11,13]. [line169-171].

 But adenomyosis is related to uterine cancers, being able to even turn into endometrial cancer or uterine sarcoma. I also published a manuscript related to this topic „Atypical Polypoid Adenomyoma of thr vagina : Follow Up and subseque Evolution: A Case Report and Update” in Diagnstics , feb 2022 PubMed, PMID:35204457. That adenomyoma of the vaginal stump ,resulted after total hysterectomy for adenomyomatous polyp with postmenopausal bleeding, underwent malignant transformation of both component, epithelial and stromal with unfavorable evolution, contrary to the results presented in the literature. I could not cite the article due to the conflict of interests, being a self-citation and also being a case report. That is why it seems important to me a comparison between the pharmaceutical products, as I have shown , there are tentative for use in the three pathologies, we hope for better results in the future.

The proposed aim of this review is to share some findings : In this review, we outline treatment options for these conditions based on their respective pathogeneses. Due to several common features, various therapeutic approaches have been explored across all three pathologies, yielding varied results.

Despite numerous drugs being tested, only a few have transitioned to clinical practice [81].[line 581-584]

Thank you very much. Your suggestions were very helpful!

Round 2

Reviewer 2 Report

Comments and Suggestions for Authors

Thank for your effort, According to in my opinion, section of endometrial cancer  should be removed out from this manuscript. I am sorry

Author Response

Dear reviewer,

I followed the editor” s instructions and I removed endometrial cancer as central part of the manuscript. We added some explanations: While endometrial cancer shares some pathogenic features with endometriosis and adenomyosis pathways, it is critical to emphasize the fundamental difference between these disease which consists in the severity of the endometrial cancer.The main concern in this case is mortality and the treatment consists of surgery with or without adjuvant therapy, personalized treatment being applied in case of recurrence or lack of response to treatment. We have to recognize this critical distinction between these pathologies before exploring various shared therapeutic options based on the complex nature of their pathogenesis. Also the treatment aim of endometrial cancer is different from endometriosis and adenomyosis. Notably, medications initially developed for endometrial cancer have found applications in treating both endometriosis and adenomyosis [10,11].This fact is based on the common features of the three pathologies at the molecular and genetic level [10,11,12,13]. The aim of this review is to identify those compounds from relevant literature. [you can find this in introduction section]

In the Results section we emphasized the important differences between endometrial cancer and the other benign pathologies, in addition to the pathogenic similarities that are the basis for the targeted treatment. I also underline in the sub-section treatments, I chose to change the title, that the idea of this review is not to describe the standardized classic treatment of endometrial cancer, but only the narrow niche of personalized treatments.

We also recognize the critical distinction in severity and health consequences between endometrial cancer and these benign diseases. The porpose of the manuscript is not to detail the standard treatment of EC for which there are already indisputable clinical guidelines, but to explore personalized treatment that targets various pathogenic pathways and may also be applicable in endometriosis or adenomyosis.  [ section:discussion]

Now I understood the major concerns related to the association I made, but I hope I managed to clarify our intention.

Besides the track change, I chose to highlight the important explanations in the manuscript in yellow.

I really hope it is ok now.

Thank you!

Round 3

Reviewer 2 Report

Comments and Suggestions for Authors

The headline at the below may be better and interesting, because your headline is emphasizing the place of endometrial cancer in this article, so much. 

 ‘Shared Pathogenic and therapeutic chacteristics of Endometriosis, Adenomyosis, and Endometrial Cancer: A Comprehensive Literature Review‘

thank for your effort

Author Response

Dear reviewer

I changed the title as you suggested.

Thank you very much!
